# Measuring Cardiorespiratory Fitness without Exercise Testing: The Development and Validation of a New Tool for Spanish Adults

**DOI:** 10.3390/jcm13082210

**Published:** 2024-04-11

**Authors:** Helmut Schröder, Isaac Subirana, Roberto Elosua, Anna Camps-Vilaró, Helena Tizón-Marcos, Montserrat Fitó, Santiago F. Gómez, Irene R. Dégano, Jaume Marrugat

**Affiliations:** 1Cardiovascular Risk and Nutrition Research Group, Hospital del Mar Medical Research Institute (IMIM), 08003 Barcelona, Spain; mfito@researchmar.net (M.F.); sgomez@gasolfoundation.org (S.F.G.); 2CIBER of Epidemiology and Public Health (CIBERESP), Instituto de Salud Carlos III (ISCIII), 28029 Madrid, Spain; 3Epidemiology and Genetics Research Group (EGEC), Hospital del Mar Research Institute (IMIM), 08003 Barcelona, Spain; isubirana@researchmar.net (I.S.);; 4CIBER of Cardiovascular Diseases (CIBERCV), Instituto de Salud Carlos III (ISCIII), 28029 Madrid, Spain; acamps@researchmar.net (A.C.-V.); htizon@psmar.cat (H.T.-M.); irene.roman@umedicina.cat (I.R.D.); 5Faculty of Medicine, University of Vic-Central University of Catalonia (UVic-UCC), 08500 Vic, Spain; 6REGICOR Study Group, Hospital del Mar Research Institute (IMIM), 08003 Barcelona, Spain; 7Biomedical Research in Heart Diseases Group, Hospital del Mar Research Institute (IMIM), 08003 Barcelona, Spain; 8Department of Cardiology, Hospital del Mar, 08003 Barcelona, Spain; 9CIBER of Pathophysiology of Obesity and Nutrition (CIBEROBN), Instituto de Salud Carlos III (ISCIII), 28029 Madrid, Spain; 10Gasol Foundation Europe, 08830 Sant Boi de Llobregat, Spain; 11Nursing and Physiotherapy Department, University of Lleida, 25198 Lleida, Spain; 12Institute for Research and Innovation in Life Sciences and Health in Central Catalonia (IRIS-CC), 08500 Vic, Spain

**Keywords:** cardiorespiratory fitness, non-exercise cardiorespiratory fitness, validity

## Abstract

**Background:** Cardiorespiratory fitness (CRF) is an important component of overall physical fitness and is associated with numerous health benefits, including a reduced risk of heart disease, diabetes, and obesity. However, direct measurement of CRF is time-consuming and therefore not feasible for screening purposes. **Methods:** A maximal treadmill exercise test with the Bruce protocol was performed to estimate VO_2max_ in 1047 Spanish men and women aged 17 to 62 years. Weight, height, and heart rate were measured. Leisure-time physical activity (LTPA) was recorded using the Minnesota Leisure Time Physical Activity Questionnaire. A multiple linear regression model was developed to predict exercise-based VO_2max_. The validity of the model was examined by correlation, concordance, Bland–Altman analysis, cross-validation, and construct validity analysis. **Results:** There was no significant difference between VO_2max_ obtained by the Bruce protocol (43.56 mL/kg/min) or predicted by the equation (43.59 mL/kg/min), with R^2^ of 0.57, and a standard error of the estimate of 7.59 mL/kg/min. Pearson’s product–moment correlation and Lin’s concordance correlation between measured and predicted CRF values were 0.75 and 0.72, respectively. Bland–Altman analysis revealed a significant proportional bias of non-exercise eCRF, overestimating unfit and underestimating highly fit individuals. However, 64.3% of participants were correctly classified into CRF tertile categories, with an important 69.9% in the unfit category. **Conclusions:** The eCRF equation was associated with several cardiovascular risk factors in the anticipated directions, indicating good construct validity. In conclusion, the non-exercise eCRF showed a reasonable validity to estimate true VO_2max_, and it may be a useful tool for screening CRF.

## 1. Introduction

Cardiorespiratory fitness (CRF) is related to the ability to perform large muscle, dynamic, moderate-to-high intensity exercise for prolonged periods. The performance of such exercise depends on the functional state of the respiratory, cardiovascular, and skeletal muscle systems [1]. Although CRF and physical activity are correlated, the former is an attribute of the latter [1]. It is well established that CRF correlates, even to a stronger degree than physical activity, with a reduced risk of premature death [2,3,4,5].

In 2016, the American Heart Association proposed considering CRF as a vital sign and to measure CRF yearly [1]. However, the determination of CRF by direct measurement of maximum oxygen uptake (VO_2max_) via ventilatory gas analysis or indirect determinations of VO_2max_ by treadmill time to exhaustion or submaximal workload is time- and resource-consuming, making it unfeasible in routine preventive clinical practice to assess cardiovascular risk. A valid alternative to the direct assessment of CRF is estimates of VO_2max_ by non-exercise test prediction equations.

The estimated CRF (eCRF) is based on easily available non-modifiable and modifiable factors related to CRF, such as sex, age, resting heart rate, anthropometric variables, and physical activity practice, which is usually obtained from questionnaires [6,7]. Self-reported physical activity, which is included in most non-exercise test eCRF equations, increases the correlation of CRF with eCRF [8]. Importantly, the accuracy of eCRF equations is generally heterogeneous, and sufficient external validity is rarely assured. It is therefore important to develop eCRF adapted to different populations [1,9]. In this context there is still no CRF equation available for the general Spanish population. Therefore, the objective of the present study was to develop and validate a brief multivariable equation to estimate CRF in Spanish adults aged 17 to 62 years when exercise data are not available.

## 2. Materials and Method

### 2.1. Study Populations

A convenience sample of 569 men and 478 women aged 17 to 62 years from the previously conducted MARATHOM (Medida de la Actividad fisica y su Relación Ambiental con Todos los lípidos en el HOMbre or Measurement of Physical Activity and its Environmental relationship with All lipids in Men) and MARATDON (Mesura de L’Activitat física i la seva Relació Ambiental amb Tots els lípids en la DONa or Measurement of Physical Activity and its Environmental relationship with All lipids in women) studies, respectively, were used for the development of the eCRF equations [10,11]. Both studies were performed to assess the amount and type of leisure-time physical activity (LTPA) advisable to keep serum lipids within low risk levels of coronary heart disease in healthy southern European men and women. Construct validity was tested in 957 participants with complete data on cardiovascular risk factors.

All participants received written information about the aims of the study in which they participated and signed an informed consent. The local Ethics Committee approved the studies on which the present project is based (94/406/I, 95/524/I, and 2016/7075/I, respectively).

### 2.2. Measurement of Indirect Maximum Oxygen Volume (VO_2max_) by Exercise-Test

All participants underwent a maximum treadmill exercise test according to the Bruce protocol [12]. Blood pressure, heart rate, and 12-lead electrocardiogram were recorded when participants were resting in the supine position. Continuous conventional 12-lead electrocardiographic monitoring was performed throughout the exercise and for 6 min post-exercise. The exercise test was maximum, and the exercise test duration was recorded. VO_2max_ was calculated as follows [13]:VO_2max_ (mL/kg/min) = 6.7 − 2.82 (men = 1, women = 2) + 0.056 (time in seconds)

### 2.3. Assessment of Leisure-Time Physical Activity

Moderate to vigorous physical activity (MVPA) was originally estimated using the Minnesota Leisure Time Physical Activity Questionnaire (MLTPA), previously validated in Spanish men and women [11,14]. To facilitate the applicability of the MLTPA, we developed and validated a short questionnaire extracted from the original [15]. We simulated the responses that would have been obtained using the latter short modification version of the MLTPA in the present study. This questionnaire includes 6 items and allows for the estimation of the total energy expenditure in LTPA, which can also be classified according to intensity (light, moderate, or vigorous). Total LTPA was measured in metabolic equivalent tasks in minutes per day (MET-min/d) and calculated as the sum of the product of frequency, duration, and intensity of each activity. LTPA levels were classified as follows: light (≤4 METs), moderate (4–5.5 METs), and vigorous (≥6 METs). MVPA included moderate and vigorous physical activity.

### 2.4. Anthropometry

Weight was measured using a calibrated precision scale. Readings were rounded up to 200 g. Height was measured in the standing position and rounded up to the nearest 0.5 cm. BMI was calculated using the standard formula of weight (kg)/height (m)^2^.

### 2.5. Cardiovascular Risk Factors

Blood samples were obtained after a 10 h fast. The serum was immediately frozen at −120 °C in liquid nitrogen for transport and stored at −80 °C for final conservation. Total cholesterol and high-density lipoprotein (HDL) cholesterol were analyzed by standardized enzymatic methods (Roche Diagnostic, Basel, Switzerland) adapted to a Cobas Mira Plus autoanalyzer (Hoffmann-La Roche, Basel, Switzerland). Smoking was self-reported by standardized questions.

### 2.6. Statistical Analysis

Quantitative variables were described as mean, standard deviation (SD), or median (intra-quartile range) if they were not normally distributed. Categorical variables were described with absolute and relative frequencies. Age was categorized into quintiles. For analytical purposes, we joined the 2nd, 3rd, and 4th quintiles. The final variable consists of three categories: 1st: 17 to 31 years, 2nd: 32 to 49 years, and 3rd: more than 49 years.

The prediction model of VO_2max_ was developed by a multiple linear regression model with the exercise-based estimation of VO_2max_ as the dependent variable and age, sex, BMI, resting heart rate, and self-reported MVPA as predictors. The coefficient of determination (R^2^) and standard error of the estimate (SEE) of the models were calculated.

The concurrent validity of the obtained regression equation was assessed by calculating Pearson’s product–moment correlation coefficients to compare the exercise test estimated VO_2max_ (reference method) with the non-exercise test estimated VO_2max_ (test method). However, it should be observed that two highly correlated measures can still show considerable differences between the two equations across their range of values. We thus calculated the concordance between the two equations by Lin’s concordance correlation coefficients (CCC) [16], the Bland–Altman plot [17] and cross-classification. The Bland–Altman method assesses the agreement between two methods by calculating the mean of their differences and regressing that figure against the average score obtained by the two methods. Proportional bias represented by possible variations in the level of agreement between methods was also analyzed to identify the possible effect of different levels of PA on the agreement between methods.

The performance of the model was further tested by leave-one-out cross-validation, a technique to avoid overfitting. The leave-one-out cross-validation evaluates the performance of a statistical model by training the model on all the data points except one and then using the model to make a prediction for the left-out point. This process is repeated for each point (1047) in the dataset, with each point being left out once. The constant error (CE) between the validation and the cross-validation sample was determined as follows:CE = Σ(measured − predicted values)/*n*

To assess construct validity, we hypothesized that the VO_2max_ derived by the non-exercise test CRF equation should be inversely associated with cardiovascular risk factors. For this purpose, sex and age adjusted general linear models were fitted to analyze the association between tertiles of non-exercise test estimated CRF and cardiovascular risk factors. Polynomial contrast was used to estimate *p* for linear trend with a post hoc Bonferroni correction for multiple comparisons.

Statistical significance was assumed when *p*-values were <0.05. Statistical analysis was performed with R software version 4.1.1.

## 3. Results

### 3.1. Study Population

The characteristics of participants in this study are shown in Table 1.

### 3.2. Prediction Function of VO_2max_

The regression equation developed to estimate non-exercise test eCRF is shown in Table 2:Estimated VO_2max_ (mL/kg/min) = 103.815 − (age × 0.261) − (sex × 10.400) − (resting heart rate × 0.163) − (BMI × 1.056) + (MVPA × 0.010)
where age is expressed in years, sex means 1 for men and 2 for women, resting heart rate is expressed in beats per minute, BMI in kg/m^2^, and MVPA in METs min/day.

All variables included in the regression model were significantly (*p* < 0.01) associated with exercise test-based estimated VO_2max_. The model explains 57.0% of the variance of exercise based estimated VO_2max_ with a SEE of 7.61 (17.5%). According to these indicators, the model performs better in men and in participants aged 31–50 years. The variance of VO_2max_ explained by the equation derived by the cross-validation sample and the corresponding %SEE were nearly the same as the ones obtained in the derivation sample (Table 2). The CE was 0 in the validation and cross-validation sample, indicating a good accuracy and internal validity of VO_2max_ estimation.

### 3.3. Concurrent Validity of the Function

The results of concurrent and absolute validity are shown in Table 3. The means of VO2max derived by the exercise and non-exercise test eCRF equations were very similar. The concurrent validity of the non-exercise test eCRF equation was good (r = 0.75 and Lin’s CCC = 0.72. The concordance between tertiles of equations, measured by cross-classification, was 64.3%, and only 2.7% were grossly misclassified. Sex and age subgroup analyses revealed no significant differences between the exercise and non-exercise test eCRF equations in men, women, and young, middle, and elderly adults. The highest correlation between the test and reference method was found in men and adults aged 32 to 49 years. Adults older than 49 years showed the highest correct cross-classification (Table 3).

### 3.4. Concordance Analysis

Table 4 shows detailed data on global and stratified concordance between the exercise and non-exercise eCRF equations determined by cross-classification. In the whole sample, the highest correct classification of non-exercise test eCRF was 69.9% in participants with low eCRF, whereas only 3.2% were found to be extremely misclassified. In stratified analyses, participants older than 49 years and women with low non-exercise eCRF showed the highest proportion of correct non-exercise test eCRF classification (84.9% and 84.3%, respectively).

### 3.5. Construct Validity

Construct validity was analyzed by correlation of the non-exercise test eCRF equation with cardiovascular risk factors (Table 5). We found a significant inverse (*p* < 0.001) association of the non-exercise test eCRF equation with total cholesterol, LDL–cholesterol, triglycerides, total cholesterol: HDL–cholesterol ratio, and triglycerides: HDL–cholesterol ratio. On the other hand, a significant direct association was observed for HDL–cholesterol.

### 3.6. Agreement between Measured and Estimated VO_2max_

The Bland–Altman plot (Figure 1) showed no significant difference between VO_2max_ derived by the non-exercise and exercise test equations, which translates in an overlapping of the reference line for 0 difference with the mean difference (0.03). However, a significant (*p* < 0.001) overestimation of VO_2max_ at low levels of CRF was found, whereas the opposite was true at high levels of CRF.

## 4. Discussion

The non-exercise test equation of CRF developed in the present study showed reasonable accuracy in predicting VO_2max_ by exercise testing in healthy Spanish men and women of a broad age range. We found no significant difference in the mean of estimated VO_2max_ derived by both measures. The non-exercise test eCRF equation overall findings remained stable in sex and age subgroup analyses with somewhat greater accuracy in men and participants aged 31–50 years. Furthermore, the non-exercise test eCRF equation showed a reasonable proportion of correctly classified participants according to their eCRF level, especially in those participants with low CRF and more than 50 years of age.

Estimated CRF is often used in population cohort studies or clinical settings as a convenient and cost-effective way to assess an individual’s fitness level, especially when direct measurement of VO_2max_ is not feasible [18,19]. The validity of estimated CRF refers to how accurately the estimate represents an individual’s true CRF level. Studies have shown that non-exercise test eCRF can be a valid predictor of true fitness level [7]. It is also important to note that ethnicity and geographic variation require local development and validation of non-exercise test estimates of CRF [20,21].

The significant predictors of measured CRF in the present study were sex, age, BMI, heart rate, and self-reported moderate and vigorous LTPA. These variables are similar to those selected in other studies on the development and validation of non-exercise eCRF equations [8,22]. These variables are easily and quickly attainable, which is especially important in time-limited clinical and epidemiological settings.

The most used measure to determine the accuracy of eCRF equations is the variance of measured CRF explained by the eCRF equation and the corresponding model error expressed as SEE and % [8,23]. In the present study, 57% of the variance in measured CRF was explained by eCRF, which is in the range reported by other studies [8]. One of the most cited eCRF equations developed by Jurca and colleagues [7] reported a variance between 58 and 65% depending on the reference method used to measure VO_2max_. The variance of explained VO_2max_ of non-exercise test eCRF functions that include self-reported physical activity ranges between 46% and 84% [8].

The mean SEE of the present prediction model was 7.61 mL/kg/min VO_2max_. This corresponds to an error of 17.5% of prediction of mean VO_2max_, within the range of most studies [8], but clearly better than non-maximal exercise testing-derived functions where it ranges from 19.1% to 27.5% [24].

Model performance was further tested by cross-validation. The coefficient of determination was almost the same in the derivation (R^2^ of 0.57) and in the cross-validation analyses (R^2^ of 0.56). In studies with external validations, the R^2^ ranged from 0.56 to 0.91 [8]. The corresponding %SEE only differed by 0.1% in the derivation and cross-validation analyses. The constant error in the derivation and cross-validation analysis was nearly zero, indicating an accurate estimation of mean VO_2max_. Furthermore, subclass analysis revealed comparable results in men and women and different age groups. Similar findings were reported from the HUNT study [18], showing a constant error close to zero in the validation and cross-validation samples and no meaningful differences in the coefficient of determination between both samples. Frequently used indicators for the validity of non-exercise derived eCRF are the correlation and concordance of the non-exercise eCRF with measured CRF. In the cross-validation analysis, we found a good correlation (r = 0.75) between measured and predicted CRF, which is in the range of existing non-exercise eCRF equations [8]. However, a good correlation between two measures does not necessarily imply a good concordance between these measures. Therefore, we tested the predictive accuracy by Lin’s concordance correlation coefficient (CCC) and Bland–Altman analysis. We found a good strength of agreement measured by Lin’s CCC (0.72), which is a robust measure of prediction and accuracy between measured and predicted CRF. Lin’s CCC value of 0.72 in the present study was somewhat higher than that found for the ACSM equation (0.68) but lower compared to the FRIEND equation (0.87) in coronary artery disease patients [25]. These differences may be drawn by differences in muscular mass, age, or even sex. Furthermore, Bland–Altman analysis revealed a significant proportional bias between measured and predicted CRF. The present eCRF equation over- and underestimates predicted VO_2max_ in the least fit and high fit individuals, respectively. This finding is in line with the findings of other studies [8] and might especially affect the correct classification of least-fit individuals [22].

One of the most important indicators of prediction accuracy, especially in clinical settings, is the correct classification of individuals in their corresponding fitness levels. In the present study, 64.3% were correctly classified into low, medium, and high fitness categories, with only 2.7% extremely misclassified. A recent review on the accuracy of CRF prediction models showed a correct classification of 52.0% on average with a range of 34.0 to 62.0% of 28 non-exercise CRF equations [22]. Evidence indicates that low-fit individuals will benefit the most from increasing their CRF [26]. Therefore, it is important that especially low-fit individuals are correctly classified into their fitness category. The present non-exercise eCRF equation correctly classified 69.9% of unfit participants in their corresponding fitness category. The capacity of correct classification of unfit individuals by the present equation is better than that found in 22 out of 28 existing non-exercise eCRF equations.

To evaluate the construct validity of the present eCRF equation, we hypothesized that the CRF derived from this equation would show a favorable relationship with cardiovascular risk factors. Indeed, it is well known that CRF is inversely associated with cardiovascular risk factors [27]. Therefore, the associations between predicted CRF and cardiovascular risk factors found in the present study confirmed the construct validity of the developed eCRF equation.

The main strength of the present study is the relatively large sample with a broad age range and the estimation of VO_2max_ by a maximal Bruce exercise test, admittedly a good method for estimating VO_2max_. The main limitation of the present study is that the maximal Bruce exercise test was performed without gas analysis. Gas analyzers allow for precise measurement of oxygen consumption and carbon dioxide production, which helps in determining exercise intensity accurately. Although the maximal Bruce exercise test is valuable for evaluating cardiovascular fitness, its effectiveness is limited when conducted without a gas analyzer, resulting in less precise measurements of VO_2max_. Therefore, the standard error of the estimate of the prediction of VO_2max_ performed by the Bruce maximal exercise test without gas analysis is considerable [28]. It is important to note that the standard error of the estimate is a measure of the accuracy of predictions, but it is not an indicator of the reliability of the test itself. Despite the drawbacks of not employing gas analysis, the Bruce maximal exercise test conducted without gas analysis can still offer valuable insights into cardiovascular fitness and can be considered a valid method to estimate VO_2max_ [29,30]. A further limitation is the simulation of responses to physical activity that would have been obtained using the validated REGICOR short physical activity questionnaire [15]. However, we found good correlations (r = 0.91) and concordance (kappa = 0.71) between simulated and original responses, recorded by the short REGICOR and MLTPA questionnaires. We deem these minor differences to be worth to facilitate the feasibility of questionnaire application in clinical practice.

## 5. Conclusions

The developed non-exercise test eCRF equation shows an acceptable accuracy and adequately predicts true VO_2max_. The reasonable correct classification of individuals, especially those with a low CRF, by the present eCRF is particularly important because this target group will benefit the most from increasing CRF. The present eCRF equation is a useful tool to predict maximal exercise test CRF, which, together with its good construct validity, may be applied in usual sex and age epidemiological and clinical settings. The prediction is particularly satisfactory for those over the age of 49. The data provided suggest that CRF could potentially be evaluated using the current non-exercise test model. However, further research is required to ascertain the practicality of this approach in primary care and other contexts, to confirm the accuracy of non-exercise CRF estimates as predictors of health outcomes, and to determine how well eCRF can detect changes in cardiorespiratory fitness over time. Finally, future studies should address the validation of this equation in other cohorts.

## Figures and Tables

**Figure 1 jcm-13-02210-f001:**
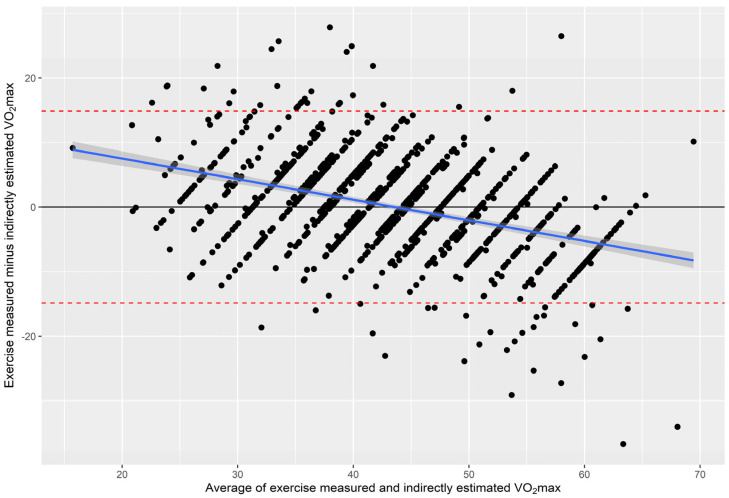
Agreement between measured and estimated VO_2max_ by Bland–Altman method. Red dashed lines: 95% confidence interval. Black line: difference between measured and estimated VO_2max_. Blue line: regression line (regression coefficient: −0.319, 95%CI −0.316–0.257, *p* < 0.001) of the association between measured and estimated VO_2max_.

**Table 1 jcm-13-02210-t001:** Characteristics of the study participants by age.

	Men *n* = 569	Women *n* = 478
Age (years)	40.1 (9.0)	39.6 (11.0)
Baseline Heart Rate (bpm)	69.5 (13.4)	74.6 (13.2)
Body mass index (kg/m^2^)	24.7 (3.05)	24.0 (3.59)
Moderate-vigorous Physical Activity (METs/min/d)	192 [41.1; 398] *	208 [89.6; 414] *
VO_2max_ (mL/kg/min)	48.0 (10.7)	38.3 (10.3)
eVO_2max_ (mL/kg/min)	48.0 (6.83)	38.3 (7.70)
Glucose (mg/dL)	86.7 (15.0)	94.5 (10.1)
Total cholesterol (mg/dL)	208 (38.9)	200 (37.7)
C-HDL (mg/dL)	49.1 (12.5)	59.6 (13.6)
C-LDL (mg/dL)	138 (37.1)	127 (34.6)
Triglycerides (mg/dL)	88.0 (64.0; 120)	63.0 (51.0; 83.0)

Mean (standard deviation) if not otherwise specified. * median (interquartile range). VO_2max_: maximal oxygen consumption, eVO_2max_: estimated maximal oxygen consumption, C-HDL: High-density lipoprotein cholesterol, C-LDL: Low-density lipoprotein cholesterol. Total cholesterol: *n* men = 477, *n* women = 468; C-HDL: *n* men = 475, *n* women = 467; C-LDL: *n* men = 472, *n* women = 463; Triglycerides: *n* men = 472, *n* women = 468; Glucose: *n* men = 477, *n* women = 468.

**Table 2 jcm-13-02210-t002:** Multiple regression equation coefficients and statistics for predicting VO_2max_.

	*β* Coefficient	95% CI	*p* Value
Validation sample			
Intercept	103.815		
Sex (1 = men; 2 = women)	−10.400	−11.360; −9.440	<0.001
Age (years)	−0.261	−0.311; −0.211	<0.001
Resting heart rate (bpm)	−0.163	−0.199; −0.127	<0.001
BMI (kg/m^2^)	−1.056	−1.211; −0.902	<0.001
MVPA (METs/min/d)	0.010	0.008; 0.012	<0.001
Model performance by sex and age groups	R^2^	SEE	%SEE
All (*n* = 1047)	0.57	7.59	17.4
Men (*n* = 569)	0.55	7.18	15.0
Women (*n* = 478)	0.41	7.91	20.6
17–31 y (*n* = 223)	0.47	7.11	14.8
32–49 y (*n* = 616)	0.53	7.62	17.1
≥50 y (*n* = 202)	0.50	7.87	22.2
Cross-validation by the leave-one-out method	R^2^	SEE	%SEE
All (*n* = 1047)	0.56	7.65	17.6
Men (*n* = 569)	0.54	7.23	15.1
Women (*n* = 478)	0.40	7.97	20.8
17–31 y (*n* = 223)	0.45	7.20	15.0
32–49 y (*n* = 616)	0.52	7.67	17.2
≥50 y (*n* = 202)	0.49	7.93	22.2

BMI = body mass index; MVPA = Moderate to vigorous physical activity; R^2^ = coefficient of determination; VO_2max_ = maximal oxygen consumption; SEE = Standard error of the estimate. %SEE calculated as (SEE * 100/mean of VO_2max_).

**Table 3 jcm-13-02210-t003:** Predictive accuracy of estimated cardiorespiratory fitness overall and by sex and age subgroups.

	All (*n* = 1047)	Men (*n* = 569)	Women (*n* = 478)	17 to 31 Years (*n* = 223)	32 to 49 Years (*n* = 616)	≥50 Years (*n* = 202)
Mean, VO_2max_, mL/kg/min (SD)						
- Exercise CRF	43.56 (11.6)	47.99 (10.66)	38.29 (10.30)	47.95 (9.71)	44.63 (11.07)	35.68 (11.33)
- Non-exercise eCRF	43.59 (8.7)	48.03 (6.83)	38.32 (7.70)	48.78 (7.11)	44.06 (7.91)	36.67 (8.08)
Difference of means, (95% CI) ^1^	−0.03 (−0.49; 0.43)	−0.03 (−0.63; 0.56)	−0.03 (−0.75; 0.68)	−0.83 (−1.77; 0.11)	0.58 (−0.03; 1.18)	−0.99 (−2.06; 0.08)
Pearson correlation coefficient	0.75	0.74	0.64	0.68	0.73	0.71
Lin’s CCC	0.72	0.67	0.62	0.65	0.69	0.67
Correct classification ^2^, (%)	64.3	66.6	61.5	62.7	61.9	73.1
Gross misclassification ^2^ (%)	2.7	2.6	2.7	1.3	3.6	1.4

Mean (standard deviation) if not otherwise specified. ^1^ Calculated as exercise eCRF-non-exercise Ecrf; ^2^ calculated across tertiles of equations; CCC: concordance correlation coefficient; CRF: cardiorespiratory fitness; CI: confidence interval SD: standard deviation; VO_2max_: maximal oxygen consumption.

**Table 4 jcm-13-02210-t004:** Agreement of cross-classification of tertiles of measured and estimated cardiorespiratory fitness.

	Measured CRF		
	Low	Medium	High
Predicted CRF			
All (*n* = 1047)			
Low (mean = 33.8 ± 4.8 VO_2_/kg/min)	**69.9**	25.4	4.8
Medium (mean = 43.8 ± 2.2 VO_2_/kg/min)	24.6	**49.9**	25.3
High (mean = 52.9 ± 4.0 VO_2_/kg/min)	3.2	23.3	**73.5**
Men (*n* = 569)			
Low (mean = 36.4 ± 3.5 VO_2_/kg/min)	**43.7**	46.0	10.3
Medium (mean = 44.2 ± 2.2 VO_2_/kg/min)	7.3	**54.8**	37.9
High (mean = 53.1 ± 3.9 VO_2_/kg/min)	0.8	13.9	**85.3**
Women (*n* = 478)			
Low (mean = 33.1 ± 4.8 VO_2_/kg/min)	**84.3**	14.0	1.7
Medium (mean = 43.4 ± 2.2 VO_2_/kg/min)	42.0	**44.9**	13.1
High (mean = 51.8 ± 4.3 VO_2_/kg/min)	12.3	57.5	**30.1**
17–31 y (*n* = 223)			
Low (mean = 36.9 ± 2.4 VO_2_/kg/min)	**35.7**	57.1	7.1
Medium (mean = 43.7 ± 2.3 VO_2_/kg/min)	7.3	**52.2**	40.6
High (mean = 54.1 ± 4.7 VO_2_/kg/min)	1.0	18.2	**80.8**
31–50 y (*n* = 688)			
Low (mean = 34.8 ± 4.3 VO_2_/kg/min)	**63.8**	28.7	7.4
Medium (mean = 44.0 ± 2.2 VO_2_/kg/min)	31.4	**48.6**	20.0
High (mean = 52.4 ± 3.4 VO_2_/kg/min)	3.7	23.4	**72.9**
>50 y (*n* = 190)			
Low (mean = 31.9 ± 5.0 VO_2_/kg/min)	**84.9**	14.4	0.7
Medium (mean = 43.2 ± 2.1 VO_2_/kg/min)	29.8	**51.1**	19.1
High (mean = 51.1 ± 3.2 VO_2_/kg/min)	9.0	45.5	**45.5**

Values are presented in percentage of concordance. Bold represents percentage of correct classified cases. CRF: cardiorespiratory fitness; VO_2_: oxygen consumption.

**Table 5 jcm-13-02210-t005:** Cardiometabolic characteristics of participants by tertiles of VO_2max_ estimated by non-exercise method (*n* = 935).

	Tertiles of VO_2max_ Non-Exercise eCRF (mL/kg/min)			
	Low (*n* = 329)	Medium (*n* = 312)	High (*n* = 294)	*p* Linear Trend
VO_2max_ (mL/kg/min) *	33.7 (20.3–39.8)	43.8 (39.9–47.3)	52.7 (47.4–66.2)	<0.001
Total cholesterol (mg/dL)	208 (204; 212)	207 (203; 211)	198 (194; 202)	0.003
LDL-Cholesterol (mg/dL)	136 (132; 140)	135 (131; 138)	126 (122; 1309	0.003
HDL-Cholesterol (mg/dL)	50.7 (49.2; 52.2)	53.6 (52.2; 55.1)	59.0 (57.4; 60.6)	<0.001
Total cholesterol:HDL-cholesterol ratio	4.5 (4.3; 4.6)	4.1 (4.0; 4.3)	3.4 (3.3; 3.6)	<0.001
Triglycerides:HDL-cholesterol ratio	2.5 (2.3; 2.7)	2.0 (1.8; 2.2)	1.1 (0.9; 1.3)	<0.001
Triglycerides (mg/dL)	107 (101; 113)	91.8 (86.4; 97.2)	64.2 (58.3; 70.1)	<0.001
Fasting blood glucose (mg/dL)	90.7 (89.2; 92.2)	89.1 (90.2; 93.4)	91.8 (90.3; 93.4)	0.355

* Median and interquartile values; all others: mean and 95% CI. HDL: high-density lipoprotein; LDL: low-density lipoprotein; eCRF: estimated cardiorespiratory fitness. VO_2max_: maximal oxygen consumption.

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
