# Peer review of "Measuring Cardiorespiratory Fitness without Exercise Testing: The Development and Validation of a New Tool for Spanish Adults"

_jcm, 2024, doi:10.3390/jcm13082210_

Round 1

Reviewer 1 Report

Comments and Suggestions for Authors

Thank you for the opportunity to review the manuscript entitled: Measuring Cardiorespiratory Fitness without Exercise Testing: Development and Validation of a new Tool for Spanish Adults. The authors have tackled an interesting topic with practical and clinical potential. I would like to ask them to consider some modifications, especially regarding text editing and descriptions of tables and graphics. 

I would like to ask to rethink the number of afiliations of the authors of this paper. The manuscript lists 9 authors and 13 affiliations. Does this mean that 13 research centers participated in the creation of this manuscript? This is an interesting ethical question. Is it necessary to indicate the acronym names of the enumerated research units? 

Please translate the notations in Spanish into English and put the Spanish-language notation in italics. 

Please remove the dots from the titles (both the title of the manuscript and the chapter's titles) 

In general, most of the titles of figures and tables are to be improved, in addition, all the information under the tables needs to be reconsidered. Mostly Figure No. 2. Each table and figure needs to meet the principle of self-description. Table captions are not the place to explain statistical analysis methods. However, they are to include explanations of all abbreviations and symbols used. 

I also ask you to think about the conclusions chapter. Please indicate the application conclusions. 

It is also worth indicating in a few sentences what are the strengths of the present project and any study limitations.

Author Response

Comment: Thank you for the opportunity to review the manuscript entitled: Measuring Cardiorespiratory Fitness without Exercise Testing: Development and Validation of a new Tool for Spanish Adults. The authors have tackled an interesting topic with practical and clinical potential. I would like to ask them to consider some modifications, especially regarding text editing and descriptions of tables and graphics. 

Reply: Thank you for this assessment our work. We have modified the manuscript according to the suggestions of the reviewer.

Comment: I would like to ask to rethink the number of afiliations of the authors of this paper. The manuscript lists 9 authors and 13 affiliations. Does this mean that 13 research centers participated in the creation of this manuscript? This is an interesting ethical question. Is it necessary to indicate the acronym names of the enumerated research units? 

Reply: We agree that this is an interesting ethical question. Currently the publishing policy of most journals, such as JCM, allows the listing of multiple affiliations. This responds to “real-world” need of researchers who belong to multiple institutions (e.g., a research center + a university + a national organization of research). In general, all institutions have contributed in different amounts to the project with salaries, support personnel, office-related expenses, lab space and yet other facilities difficult to list exhaustively. The main point in our vision is to ensure that all co-authors comply with authorship international rules, which is the case here. We feel inclined to respect all co-authors wishes of acknowledging their institutional support. Maybe the editors of the journal may be willing to comment on this point.

Comment: Please translate the notations in Spanish into English and put the Spanish-language notation in italics. 

Reply: Thank you for noting this. We have done as requested on page 2 line 75 to 81: “…conducted MARATHOM (Medida de la Actividad fisica y su Relación Ambiental con Todos los lípidos en el HOMbre or Measurement of Physical Activity and its Environmental relationship with All lipids in Men) and MARATDON (Mesura de L’Activitat física i la seva Relació Ambiental amb Tots els lípids en la DONa or Measurement of Physical  Ativity and its Environmental relationship with All lipids in women) studies, respectively,...”.  (page 2,  lines 78-83 )

Comment: Please remove the dots from the titles (both the title of the manuscript and the chapter's titles).

Reply: The title reads now “Measuring Cardiorespiratory Fitness without Exercise Testing. Development and Validation of a new Tool for Spanish Adults”. Additionally, we have removed the dots from the chapter’s titles.

Comment: In general, most of the titles of figures and tables are to be improved, in addition, all the information under the tables needs to be reconsidered. Mostly Figure No. 2. Each table and figure needs to meet the principle of self-description.

Reply: We have rewritten the titles of the table as follows:

Table 1

Previous: Characteristics of the study population.

Current: Characteristics of the study participants by sex.

Table 2

Previous:  Linear regression coefficients to predict exercise test VO2max with non-exercise cardiorespiratory fitness function estimation, and R2 in sex and age subgroups in the whole population and in a leave-one-out cross-validation.

Current: Multiple regression equation coefficients and statistics for predicting VO2max

Table 3

Previous: Comparison of VO2max by exercise test and estimated by cardiorespiratory fitness equation, correlation coefficient and agreement between exercise test and estimated of VO2max in 1047 participants, overall and in sex and age subgroups. 

Current: Predictive accuracy of non-exercise estimated cardiorespiratory fitness

Table 4

Previous: Cross-classification of measured and estimated cardiorespiratory fitness (CRF)

Current: Agreement of cross-classification of tertiles of measured and estimated cardiorespiratory fitness (CRF). In bold percentage of correctly classified cases.

Table 5

Previous: Cardiometabolic profile of participants according to classification of  non-exercise estimated VO2max (n=935).

Current: Cardiometabolic characteristics of participants by tertiles of VO2max estimated by non-exercise method (n=935)

Figure 1:  We regret this error. We have included the following title for figure 1: “Agreement between measured and estimated maximal oxygen consumption (VO2max) by Bland Altman method”. Additionally, we added the following caption: “

Red dashed lines: 95% confidence interval

Black line: Difference between measured and estimated VO2max

Blue line: Regression line (regression coefficient: -0.319, 95%CI -0.316-0.257, p<0.001) of the association between measured and estimated VO2max

Comment: Table captions are not the place to explain statistical analysis methods. However, they are to include explanations of all abbreviations and symbols used. 

Reply: We have modified tables captions accordingly.

Comment: I also ask you to think about the conclusions chapter. Please indicate the application conclusions. 

Reply: We have included the following text in the conclusion chapter: “The data provided suggests that CRF could potentially be evaluated using the current non-exercise test model. However, further research is required to ascertain the practicality of this approach in primary care and other contexts, to confirm the accuracy of non-exercise CRF estimates as predictors of health outcomes, and to determine how well eCRF can detect changes in cardiorespiratory fitness over time. Finally, future studies should address the validation of this equation in other cohorts.”  (page 10, lines 433-439).

Comment: It is also worth indicating in a few sentences what are the strengths of the present project and any study limitations.

Reply: The strength and limitation section of the discussion reads as follows: “The main strength of the present study is the relatively large sample with a broad age range and the estimation of VO2max by a maximal Bruce exercise test admittedly a good method to estimate VO2max. The main limitation of the present study is that the maximal Bruce exercise test was performed without gas analysis. Gas analyzers allow for precise measurement of oxygen consumption and carbon dioxide production, which helps in determining exercise intensity accurately. Although the maximal Bruce exercise test is valuable for evaluating cardiovascular fitness, its effectiveness is limited when conducted without a gas analyzer, resulting in less precise measurements of VO2max.  Therefore, the standard error of the estimate of the prediction of VO2max performed by the Bruce maximal exercise test without gas analysis is considerable [29]. It is important to note that the standard error of the estimate is a measure for the accuracy of predictions, but it is not an indicator for the reliability of the test itself. Despite the drawbacks of not employing gas analysis, the Bruce maximal exercise test conducted without gas analysis can still offer valuable insights into cardiovascular fitness  and can be considered a valid method to estimate VO2max [30,31]. A further limitation is the simulation of responses of physical activity that would have been obtained using the validated REGICOR short physical activity questionnaire [15]. However, we found good correlations (r=0.91) and concordance (kappa=0.71) be-tween simulated and original responses, recorded by the short REGICOR and MLTPA questionnaires. We deem these minor differences to are worth to facilitate the feasibility of questionnaire application in clinical practice.” (page: 9 and 10, lines 406-425).

Reviewer 2 Report

Comments and Suggestions for Authors

The authors proposed a multiple linear regression model to estimate the cardiorespiratory fitness in Spanish men and woman based on age, sex, body mass index, basal heart rate and leisure time physical activity. The model main strength is the capacity of generalization in a broad age range, in both man and woman, with a good prediction (non-exercise estimated VO2max), moderate reliability (Pearson correlation, Lin’s and Bland-Altman concordance with exercise estimated VO2max), moderate sensitivity and sensibility (Cross-classification) and good validity (Cardiovascular risk factors). On the other hand, the main limitation of this approach is the concurrent validity through an indirect method. Therefore, the error of the estimate present in the reference method may account to the error of the estimate of the test method and the real error of the estimate is unknown.

Minor comments

The introduction can benefit of contextualization for the novelty of the model. Moreover, the discussion can benefit of future directions for improving the model (e.g. a more specific model to age and sex) and suggestions (e.g. to verify the model sensibility to track longitudinal changes).

Major comments

In the discussion, I believe it is essential to inform the standard error of the estimate or equivalent error present in the Bruce protocol and any relevant information regarding the concurrent validity of the reference test. Further, readers should be aware of the potential bias of the model and that the estimate of the cardiorespiratory fitness by the equation must be interpreted with caution.

Author Response

The authors proposed a multiple linear regression model to estimate the cardiorespiratory fitness in Spanish men and woman based on age, sex, body mass index, basal heart rate and leisure time physical activity. The model main strength is the capacity of generalization in a broad age range, in both man and woman, with a good prediction (non-exercise estimated VO2max), moderate reliability (Pearson correlation, Lin’s and Bland-Altman concordance with exercise estimated VO2max), moderate sensitivity and sensibility (Cross-classification) and good validity (Cardiovascular risk factors). On the other hand, the main limitation of this approach is the concurrent validity through an indirect method. Therefore, the error of the estimate present in the reference method may account to the error of the estimate of the test method and the real error of the estimate is unknown.

Reply: We agree with the reviewer that the main limitation of the present study is the choice of the reference method. We have addressed this topic in the limitation section of the manuscript.

Minor comments

The introduction can benefit of contextualization for the novelty of the model. Moreover, the discussion can benefit of future directions for improving the model (e.g. a more specific model to age and sex) and suggestions (e.g. to verify the model sensibility to track longitudinal changes).

Reply: We have modified the text in the introduction as follows: “Importantly, the accuracy of eCRF equations is generally heterogeneous, and sufficient external validity is rarely assured. It is therefore important to develop eCRF adapted to different populations [1,9]. In this context there is still no CRF equation available for the general Spanish population. Therefore, the objective of the present study was to develop and validate a brief multivariable equation to estimate CRF in Spanish adults aged 17 to 62 years when exercise data is not available. (page 2, lines:  68-74).

Additionally, we have added the following text to address in the discussion future directions: “However, further research is required to ascertain the practicality of this approach in primary care and other contexts, to confirm the accuracy of non-exercise CRF estimates as predictors of health outcomes, and to determine how well eCRF can detect changes in cardiorespiratory fitness over time. Finally, future studies should address the validation of this equation in other cohorts”. (page 10, lines 433-439).

Major comments

In the discussion, I believe it is essential to inform the standard error of the estimate or equivalent error present in the Bruce protocol and any relevant information regarding the concurrent validity of the reference test. Further, readers should be aware of the potential bias of the model and that the estimate of the cardiorespiratory fitness by the equation must be interpreted with caution.

Reply: We have added the following text to the discussion section of the manuscript according to the suggestion of the reviewer: “The main limitation of the present study is that the maximal Bruce exercise test was performed without gas analysis. Gas analyzers allow for precise measurement of oxygen consumption and carbon dioxide production, which helps in determining exercise intensity accurately. Although the maximal Bruce exercise test is valuable for evaluating cardiovascular fitness, its effectiveness is limited when conducted without a gas analyzer, resulting in less precise measurements of VO2max.  Therefore, the standard error of the estimate of the prediction of VO2max performed by the Bruce maximal exercise test without gas analysis is considerable [29]. It is important to note that the standard error of the estimate is a measure for the accuracy of predictions, but it is not an indicator for the reliability of the test itself. Despite the drawbacks of not employing gas analysis, the Bruce maximal exercise test conducted without gas analysis can still offer valuable insights into cardiovascular fitness  and can be considered a valid method to estimate VO2max [30,31].“ (page 10,  lines 408-425).